behaviour/evolution/health and disease and epidemiology

menopause, sexual frequency, trade-off, pheromones, cox regression, Study of Women's Health Across the Nation

**Author for correspondence:**
Megan Arnot
e-mail: megan.arnot.13@ucl.ac.uk

# Sexual frequency is associated with age of natural menopause: results from the Study of Women's Health Across the Nation

## Megan Arnot[1] and Ruth Mace[1]

[1]Department of Anthropology, University College London, 14 Taviton Street, London WC1H 0BW, UK

(iD) MA, 0000-0002-6293-5202; RM, 0000-0002-6137-7739

It is often observed that married women have a later age of natural menopause (ANM) than unmarried women; however, the reason for this association is unknown. We test an original hypothesis that sexual frequency acts as a bio-behavioural mediator between marital status and ANM. We hypothesize that there is a trade-off between continued ovulation and menopause based on the woman's chances of becoming pregnant. If a woman is sexually inactive, then pregnancy is impossible, and continued investment in ovulation would not be adaptive. In addition, we test an existing hypothesis that the observed relationship is because of the exposure to male pheromones. Data from 2936 women were drawn from 11 waves of the Study of Women's Health Across the Nation, which is a longitudinal study conducted in the United States. Using time-varying Cox regression, we found no evidence for the pheromone hypothesis. However, we did observe that women who reported to have sex weekly during the study period were 28% less likely to experience menopause than women who had sex less than monthly. This is an indication that ANM may be somewhat facultative in response to the likelihood of pregnancy.

## 1. Background

There is a great deal of cross-cultural variation in age of natural menopause (ANM) [1]; and while it is largely governed by genetic factors [2], it is estimated that up to half of the population variance in menopause timing is the result of non-genetic influences [3]. The number of follicles a woman has is established *in utero* at around five months gestation, and is therefore finite at approximately

seven million [4]. By the time a woman reaches puberty, this number has already declined to approximately 400 000 [5], with menopause occurring once the ovarian reserve has dropped to below 1000 [6]. While integrally a biological process, a number of behavioural and lifestyle factors have been found to associate with ANM, including smoking habits [7–20] and educational attainment (which is used as an indicator of socioeconomic status) [9–11].

One puzzling association reported in epidemiological literature is the relationship between marital status and menopause timing, in which married women attain menopause later than never married or divorced women [7,8,12,14,21–23], but there is little understanding of the pathway connecting the two. One existing hypothesis postulates that the relationship between ANM and marital status is the result of male–female cohabitation [7]. This is based on the idea that increased exposure to male pheromones (as a result of being married, and therefore cohabiting) may increase the likelihood of having a regular menstrual cycle [24], with regular menstrual cycles having been observed to delay the menopause [25]. As an alternative, we propose an adaptive explanation based on an energetic trade-off. As married people typically have sex more often than those who are uncoupled [26,27], we suggest that the observed relationship between marital status and ANM may be capturing the effect of sexual frequency during the pre- and peri-menopause on menopause timing. Ovulation can be seen as a costly process, both in terms of energetics and owing to its impairing effect on the immune system [28,29]. As a result, should a woman be having little or infrequent sex when approaching midlife, then the body will not be receiving the physical cues of a possible pregnancy, and it may therefore not be adaptive to invest resources into continued ovulation. Rather, it would be better from a fitness-maximizing perspective for the woman to cease fertility and invest energy into any existing kin she has [30–32]. Conversely, if the woman is still engaging in sex regularly, then it may be adaptive for her to continue ovulating for slightly longer, allowing her to increase her direct fitness.

## 1.1. Aims and hypotheses

Currently, there is no clear reason why married women experience a later menopause, and we propose a functional reason for this relationship. Here, we test the two following hypotheses:

(i) H1: increased sexual frequency lowers the risk of entering menopause; and
(ii) H2: exposure to male pheromones delays menopause.

We acknowledge that this study is largely correlative, but nonetheless, if positive results are found in favour of H1, then we would suggest that menopause timing could be facultative and that the within- and between-population differences in ANM are somewhat adaptive.

# 2. Material and methods

## 2.1. Study sample

Data from 2936 women were drawn from the Study of Women's Health Across the Nation (SWAN), which is an on-going community-based, multi-site, longitudinal cohort study currently being carried out in the United States of America, specifically designed to collect data on the biological and psychosocial changes that occur alongside the menopause. Despite being a community-based sample, SWAN is thought to be the largest, most diverse and most representative study currently available to research aspects of the menopausal transition [33]. Criteria to be part of the baseline cohort (recruited in 1996/1997) included being aged between 42 and 52, having an intact uterus, at least one ovary, not being pregnant, having experienced a menstrual cycle within the past three months, and self-identifying as one of the five pre-specified racial/ethnic groups (African-American, Chinese or Chinese American, Japanese or Japanese American, non-Hispanic Caucasian or Caucasian) [34]. The current analysis uses data from the baseline interview and 10 follow-up visits (1996–2007), which are publicly available online [35–46].

## 2.2. Variables

### 2.2.1. Age of menopause

Menopause timing was the primary variable of interest within the study. Biomedically, a woman is defined as having experienced menopause once she has experienced 12 months of amenorrhoea in the

absence of external influence over menstruation (e.g. breastfeeding, hormonal contraceptives) [47]. SWAN provides an existing derived variable that conforms to this definition, and this was used to code women as having experienced menopause, not entered menopause (e.g. still menstruating, peri-menopausal), or ceased menstruation for another reason (e.g. hysterectomy).

### 2.2.2. Sexual frequency

To test the first hypothesis, we looked at ANM in relation to sexual frequency. A time-varying 'sex index' was derived from the woman's responses to questions about her sex habits, which included:

(i) 'during the past 6 months, have you engaged in sexual activities with a partner?' (yes; no);
(ii) 'during the past 6 months, how often, on average, have you engaged in each of the following activities: sexual touching or caressing; oral sex; sexual intercourse?' (not at all; once or twice a month; about once a week; more than once a week; daily); and
(iii) 'on average, in the last 6 months, how often have you engaged in masturbation (self-stimulation)?' (not at all; less than once a month; once or twice a month; about once a week; more than once a week; daily).

For each woman, the maximum amount of sexual activity from any of the aforementioned questions was taken as being her sexual frequency. For example, if a woman reported having intercourse 'once or twice a month', but oral sex 'daily', 'daily' was recorded as her sex index. Sexual activity other than intercourse was used to create the sex index as the hypothesis is predicting that cues from sex will result in a trade-off, and the underlying mechanism of sexual touching, oral sex and masturbation could all signal possible pregnancy to the body. Owing to the small amount of responses in some of the categories they were aggregated into three new categories, with 'less than monthly' comprising of women who reported to not have had any sex in the past six months, as well as those who responded 'not at all' or 'less than once a month' within the sexual activities questions. 'Monthly' sex was used to code women who have engaged in any form of sexual activity 'once or twice a month'; and 'weekly' if a women reports engaging in any form of sexual activity 'about once a week', 'more than once a week' or 'daily'.

### 2.2.3. Male pheromones

As there is no direct measure of male pheromone exposure, we use male household presence as a proxy of male pheromones. Three variables were derived from SWANs questions regarding household composition. Firstly, a binary variable was created based on whether the respondent reported living with a romantic male partner, such as a husband, boyfriend, fiancé or similar. A second binary variable was made which then coded whether the woman lived with any male (e.g. husband, son, male friend), and this was also included as a continuous variable that counted the number of males living in the house.

### 2.2.4. Covariates

Marital status was included as a covariate when looking at the relationship between sexual frequency and ANM, based on previous literature isolating it as being associated with menopause timing. It was not included when testing the pheromone hypothesis owing to the high degree of collinearity with male household presence. To create this variable, women's responses to questions regarding their relationship status and living arrangements were used, with women subsequently being coded as 'divorced, separated or single', 'married or in a relationship', or 'widowed'.

Additional variables were selected based on existing research looking at what influences ANM. Time-varying covariates included the woman's smoking habits (never smoked; ever smoked) [7–20] and body mass index (BMI) [10,11,14,16,17,19–21,48,49]. Race (self-identified as black or African American; Chinese; Japanese; non-Hispanic Caucasian; Hispanic) [12–14,18,50], educational attainment (less than high school education; high school education; some college/technical school; college degree; post-graduate education) [9–12,14,18–22,48], parity [7,9–12,14,15,18–22,48,51,52] and age of menarche [9,10,17,19,20,51,53] were also included as time-invariant covariates. Serum oestradiol levels and a measure of self-perceived overall health were also included as time-varying covariates to adjust for the health and hormonal changes that occur throughout the menopausal transition, which can also affect a woman's likelihood and desire to engage in sex [54]. Furthermore, oestradiol plays a role in the mechanism of releasing eggs and also has a bidirectional role relationship with sexual frequency, as oestradiol levels have also been found to influence the sex drive, and sexual interactions themselves can increase oestradiol levels [55,56].

## 2.3. Analyses

Time-varying Cox proportional hazards modelling was used to conduct an event history analysis, which is a powerful method of regression modelling that allows the isolation of precise effects over the risk of an event (i.e. the menopause) happening [57]. Unlike standard regression models, it is able to deal with time-series and censored data and produces a hazard ratio (HR), which is a measure of the risk of the event happening. The age of the participant was used as the time-scale in the models, rather than time since first interview [58]. Analyses were carried out in R (version 3.5.3) [59] using the packages *survival* [60], and *survminer* [61].

## 2.4. Model selection

A base candidate model was created which included all the aforementioned covariates (oestradiol, education, BMI, race, smoking habits, parity, age of menarche, overall health, age at first interview), with the variables of interest being subsequently added to the models (see below, models *b–g*). We use the Akaike information criterion (AIC) to compare model fit, in which the model with the lowest AIC value best fits the data. The models were then weighted based on how much their AIC value increased compared to the best fitting model, with a decrease in AIC value of two or greater implying a better model fit [62]. Sexual frequency and marital status are not included in models with household composition owing to the high degree of collinearity. Similarly, the pheromone variables are all used in separate models as each of them capture similar information about household composition. The following candidate models were compared to test H1:

*a*, base;
*b*, base + marital status;
*c*, base + sexual frequency;
*d*, base + sexual frequency + marital status;

and the models listed below compared to test H2:

*a*, base;
*e*, base + whether the woman lives with a male partner;
*f*, base + whether the woman lives with a male;
*g*, base + total number of males living in household.

# 3. Results

## 3.1. Descriptive statistics

Mean age at first interview was 45.88 (standard deviation (s.d.): 2.70). Owing to the requirements to be part of SWAN, no one had yet entered the menopause, but 46% were in early peri-menopause, and 54% were pre-menopausal. Across the 10 years of follow-up interviews used within this study, 1324 (45%) of the 2936 women experienced a natural menopause at an average age of 52 (s.d.: 2.59).

At the entry to the study, most women were either married or in a relationship (78%), and 68% of women lived with their partner. The most frequent pattern of sexual activity was weekly (64%), and there was a wide range of oestradiol readings, with an average level of 55.05 pg/mL (interquartile range (IQR): 33–89). Non-Hispanic Caucasian women were most represented within the sample as 48% of women identified as being such, in addition to the majority of women being educated to above a high school level (some college/technical school: 32%, college degree: 20%, post-graduate education: 23%). The median BMI was 27 (IQR: 23–32), with 59% being overweight or obese, and most women reported to having never smoked regularly (57%). Women on average had two children (s.d.: 1.42) and experienced menarche at the age of 13 (s.d.: 1.67). We present full descriptive statistics of the baseline cohort in the electronic supplementary material, table S1.

## 3.2. Hypothesis 1: increased sexual frequency lowers the risk of entering menopause

Unadjusted Cox models testing the relationship between ANM and sexual frequency suggest that women who have more frequent sex during the pre- and peri-menopause have a lower risk of entering

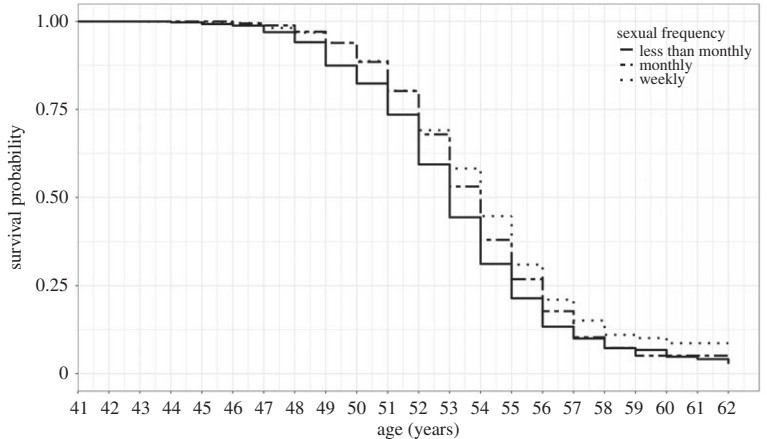

**Figure 1.** Survival curves for age at natural menopause by sexual frequency; Kaplan–Meier estimates.

menopause, with women engaging in sexual activity weekly being at the lowest risk (HR: 0.74, 95% confidence interval (CI): 0.65–0.84) (see the electronic supplementary material, table S2 and figure 1). The observation that married women enter menopause later was not replicated in this study, as unadjusted models indicted that marital status had no impact on the risk of entering menopause (married/in a relationship HR: 0.95, 95% CI: 0.84–1.08; widowed HR: 1.13, 95% CI: 0.83–1.54). Following full adjustment in model *d* the relationship between sexual frequency and ANM was maintained (sex monthly HR: 0.81, 95% CI: 0.69–0.95; sex weekly HR: 0.72, 95% CI: 0.62–0.83); however, there is now a suggestion that married women have an increased risk of entering menopause (HR: 1.19, 95% CI: 1.02–1.38) (electronic supplementary material, table S3).

### 3.3. Hypothesis 2: exposure to male pheromones delays age of natural menopause

To test whether exposure to male pheromones delay ANM, we looked at male–female cohabitation as a proxy. In line with the previous relationship found between marital status and ANM, prior to adjustment, living with a male partner was not associated with ANM (HR: 0.95, 95% CI: 0.85–1.06), with similar results being found following adjustment in model *e* (HR: 1.03, 95% CI: 0.91–1.16). Furthermore, simply having a male in the household also appears to have no relationship with ANM, regardless of whether this was modelled as looking at whether a male was present in the household or not (model *f* HR: 1.10, 95% CI: 0.95–1.27), or if it looked at the number of males in the household (model *g* HR: 1.03, 95% CI: 0.97–1.09) (electronic supplementary material, tables S2 and S3).

### 3.4. Model selection

We present model selection results in table 1. In support of previous results suggesting that male pheromones have no influence over ANM, when comparing models *a*, *e*, *f* and *g*, the base model *a* which included none of the household composition variables was found to best fit the data.

When comparing all models *a–g* and just the models testing the sexual frequency hypothesis *a–d*, models *c* and *d* were found to best fit the data, both of which include sexual frequency, adding further support to the notion that ANM is associated with sexual frequency. Results from the best model *d*, are presented in figure 2, where it can be seen that sexual frequency has a near-linear relationship with ANM, with increased sexual frequency lowering the risk of entering menopause, even when controlling for marital status, oestradiol levels, overall health and other aforementioned covariates.

## 4. Discussion

While women often stop reproducing many years prior to the menopause [63], the permanent reproductive cessation resulting from the menopause means a woman is no longer physically able to increase her direct fitness. Many lifestyle factors have been found to associate with ANM, and these are seldom discussed from an evolutionary perspective. Here, we have focused specifically on the

**Table 1.** Model selection. (Base model comprises of oestradiol, education, race/ethnicity, body mass index, smoking habits, age of menarche, overall health and age of first interview. MS, marital status; SF, sexual frequency; MP, lives with a male partner; M, lives with a male; TM, total number of males in household. $K$, number of parameters; AIC, Akaike information criterion; $w_i$, model probability. Best fitting model(s) in italics. ($n = 2936$, $n$ events $= 1324$).)

| model | $K$ | all models | | | hypothesis 1 (models $a$–$d$) | | | hypothesis 2 (models $a$, $e$–$g$) | | |
|---|---|---|---|---|---|---|---|---|---|---|
| | | AIC | ΔAIC | $w_i$ | AIC | ΔAIC | $w_i$ | AIC | ΔAIC | $w_i$ |
| $a$, base | 18 | 17 399.57 | 11.40 | 0.00 | 17 399.57 | 11.40 | 0.00 | 17 399.57 | 0.00 | 0.35 |
| $b$, base + MS | 20 | 17 403.05 | 14.88 | 0.00 | 17 403.05 | 14.88 | 0.00 | — | — | — |
| $c$, base + SF | 20 | 17 389.35 | 1.18 | 0.35 | 17 389.35 | 1.18 | 0.36 | — | — | — |
| *d*, base + SF + MS | *22* | *17 388.17* | *0.00* | *0.64* | *17 388.17* | *0.00* | *0.64* | — | — | — |
| $e$, base + MP | 19 | 17 401.37 | 13.21 | 0.00 | — | — | — | 17 401.37 | 1.81 | 0.14 |
| $f$, base + M | 19 | 17 399.88 | 11.71 | 0.00 | — | — | — | 17 399.88 | 0.32 | 0.30 |
| $g$, base + TM | 19 | 17 400.53 | 12.36 | 0.00 | — | — | — | 17 400.53 | 0.96 | 0.22 |

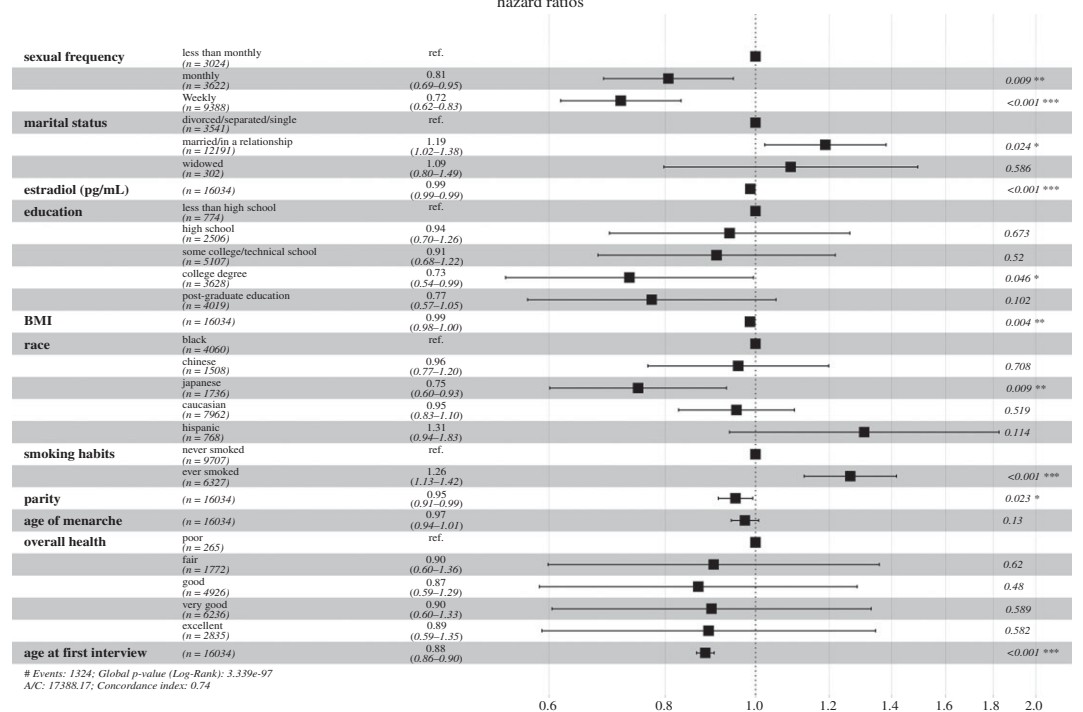

**Figure 2.** Forest plot showing the hazard ratios and 95% confidence intervals from the best fitting Cox model *d*. A lower hazard ratio indicates a decreased risk of entering menopause.

relationship between marital status and ANM, testing an original hypothesis that sexual frequency acts as a bio-behavioural mediator between ANM and marital status, in addition to the existing hypothesis that married women enter menopause later owing to male pheromones [7]. To test the latter hypothesis, three measures of male–female cohabitation were used as a proxy of exposure to male pheromones, and we found no evidence to suggest that menopause timing was responsive to living with a male, and therefore possibly male pheromones. It should be noted that this hypothesis could be fundamentally flawed, as there is no conclusive evidence either that humans produce pheromones, or that they are capable of detecting them [64]. Nonetheless, this is, to our knowledge, the first study addressing the pheromonal hypothesis since it was originally proposed, and while male household presence is merely a proxy of pheromones—and it may be that the hypothesis is moot owing to the absence of evidence for human pheromones—it is an indication that the relationship between marital status and ANM is not capturing the effect of male pheromones on the menstrual cycle.

This study did not replicate findings from previous research that married women enter menopause later. In fact, following complete adjustment, the converse was found, with women who were married or in a relationship having an increased risk of entering menopause compared to divorced, separated and single women. Conflicting results regarding marital status' effect on ANM have been found elsewhere (e.g. [65]), and one reason for this may be the way in which the researcher chooses to code the variable. In this analysis, romantic partnerships that may not have been acknowledged in previous studies owing to having not been formalized by a marriage ceremony (e.g. cohabiting but unmarried) were taken into account. In addition, some prior studies have not included marital status as time-variant and dichotomized the variable as 'ever married' or 'never married' (e.g. [66]). Hence, the responsive way in which this study coded marital status may account for the difference in results.

Another reason for this difference may be the cultural setting of previous studies. For example, research originating from Iran found that ever married women experience a later menopause than those who never married [8]. However, in the case of Iran where dowry is still common practice, it means marriage is contingent upon family wealth [67]. Therefore, the effect of marital status on ANM would be confounded by a woman's socioeconomic position, which itself would relate to other aspects of her health and life history that have been associated with menopause timing—such as BMI and age of menarche—therefore resulting in a significant difference in ANM between those who have and have not been married. Within Iran, sex outside of marriage is prohibited both legally and socially, meaning marital status would be highly correlated with sexual behaviour [68]. Hence, it may be that previous

studies identifying married women enter menopause later are simply capturing the effect of health and lifestyle patterns that themselves associate with both marital status and menopause timing, rather than demonstrating that marital status itself is a cross-cultural correlate of ANM. Future research should aim to address the cultural setting in which the data were collected when interpreting the results.

Evidence supporting the notion that ANM associated with sexual frequency during the pre- and peri-menopause was found. Even following complete adjustment, results still indicated that women who engage in sexual activity weekly or monthly have a lower risk of entering menopause relative to those who report having some form of sex less than monthly. If we interpret these results from a fitness-maximizing framework, it may be the physical cues of sex signal to the body that there is a possibility of becoming pregnant, and therefore an adaptive trade-off may occur between continued energetic investment in ovulation and reproductive cessation. During ovulation, the woman's immune function is impaired making the body more susceptible to disease [28,29]. Hence, if a pregnancy is unlikely owing to a lack of sexual activity, then it would not be beneficial to allocate energy to a costly process, especially if there is the option to invest resources into existing kin [30,31]. The idea that women cease fertility in order to invest in kin is known as the Grandmother Hypothesis, which predicts that the menopause originally evolved in humans to reduce reproductive conflict between different generations of females, and allow women to increase their inclusive fitness through investing in their grandchildren [30,69,70]. It may be costly for a woman to cease ovulatory function if the chances of her becoming pregnant are still high. In other words, if she is still able to increase her direct fitness, then it may be better to maintain the function of her menstrual cycle for slightly longer.

It should be noted that there may be a bidirectional relationship between the physical condition of the woman when approaching the menopause and sexual engagement. As oestrogen levels decline, women are more likely to experience vaginal dryness and discomfort, making them less inclined to engage in sex [71]. This study has attempted to control for this factor through adjusting for both oestradiol levels and the woman's self-perceived overall health, with the association between sexual frequency and ANM still persisting following this adjustment. This suggests that—even when controlling for the complicated relationship between health, hormonal fluctuations and desire for sex—the menopause may be somewhat facultative in response to sexual behaviour, rather than being solely the result of a physiological constraint (e.g. degrading oocyte quality).

# 5. Conclusion

This paper demonstrates that sexual frequency is associated with ANM and is also, to our knowledge, the first formal test of the hypothesis that male pheromones have an influence over ANM. While only a proxy of male pheromones was used, no association between male cohabitation and menopause timing was found, indicating that male–female cohabitation is not the driving force behind the relationship between marital status and menopause timing and that pheromones probably do not influence ANM. We did not replicate the findings from previous research showing that simply being married is associated with a later ANM, most likely owing to the variable cultural and temporal settings of previous studies. However, we did demonstrate that increased sexual frequency during the pre- and peri-menopause decreased the risk of experiencing menopause. While causation cannot be conclusively inferred, we hypothesize that this relationship is the result of an adaptive trade-off relative to the likelihood of pregnancy when approaching the menopause. Of course, the menopause is an inevitability for women, and there is no behavioural intervention that will prevent reproductive cessation; nonetheless, these results are an initial indication that menopause timing may be adaptive in response to sexual behaviour.

Ethics. Institutional Review Board approval at each site was obtained, and all women provided written consent.
Data accessibility. The data used in this study are publicly available at: https://www.icpsr.umich.edu/icpsrweb/ICPSR/series/00253. Code used to conduct analyses has been uploaded as part of the electronic supplementary material.
Authors' contributions. M.A. and R.M. designed the study. M.A. conducted the analysis and wrote the manuscript. R.M. reviewed and revised multiple drafts of the manuscript.
Competing interests. We declare we have no competing interests.
Funding. M.A. is funded by the ESRC-BBSRC Soc-B Centre for Doctoral Training (grant no. ES/P000347/1). The Study of Women's Health Across the Nation (SWAN) has grant support from the National Institutes of Health (NIH), DHHS, through the National Institute on Aging (NIA), the National Institute of Nursing Research (NINR) and the NIH Office of Research on Women's Health (ORWH) (grant nos. U01NR004061; U01AG012505, U01AG012535, U01AG012531, U01AG012539, U01AG012546, U01AG012553, U01AG012554, U01AG012495). The content of this article is solely the responsibility of the authors and does not necessarily represent the official views of the NIA, NINR, ORWH or the NIH.

Acknowledgements. We thank E. Emmott, S. Jivraj, L. Kretschmer, M. Dyble, G. D. Salali, and H. Zhang for helpful feedback at various stages of the preparation of this manuscript, in addition to two anonymous reviewers for their valuable comments. The authors are grateful to the study stuff at each site and all the women who participated in SWAN. *Clinical Centres*: University of Michigan, Ann Arbor—Siobán Harlow, PI 2011–present, Mary Fran Sowers, PI 1994–2011; Massachusetts General Hospital, Boston, MA—Joel Finkelstein, PI 1999–present; Robert Neer, PI 1994–1999; Rush University, Rush University Medical Center, Chicago, IL—Howard Kravitz, PI 2009–present; Lynda Powell, PI 1994–2009; University of California, Davis/Kaiser—Ellen Gold, PI; University of California, Los Angeles—Gail Greendale, PI; Albert Einstein College of Medicine, Bronx, NY—Carol Derby, PI 2011–present, Rachel Wildman, PI 2010–2011; Nanette Santoro, PI 2004–2010; University of Medicine and Dentistry—New Jersey Medical School, Newark—Gerson Weiss, PI 1994–2004; and the University of Pittsburgh, Pittsburgh, PA—Karen Matthews, PI. *NIH Program Office*: National Institute on Aging, Bethesda, MD—Chanda Dutta 2016–present; Winifred Rossi 2012–2016; Sherry Sherman 1994–2012; Marcia Ory 1994–2001; National Institute of Nursing Research, Bethesda, MD—Program Officers. *Central Laboratory*: University of Michigan, Ann Arbor—Daniel McConnell (Central Ligand Assay Satellite Services). *Coordinating Centre*: University of Pittsburgh, Pittsburgh, PA—Maria Mori Brooks, PI 2012–present; Kim Sutton-Tyrrell, PI 2001–2012; New England Research Institutes, Watertown, MA—Sonja McKinlay, PI 1995–2001. *Steering Committee*: Susan Johnson, Current Chair, Chris Gallagher, Former Chair.

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
