## [Reviewer comments · Royal Society Open Science]

Review History

RSOS-191020.R0 (Original submission)

Review form: Reviewer 1

Is the manuscript scientifically sound in its present form?

Yes

Are the interpretations and conclusions justified by the results?

Yes

Is the language acceptable?

Yes

Do you have any ethical concerns with this paper?

No

Have you any concerns about statistical analyses in this paper?

No

Recommendation?

Accept with minor revision (please list in comments)

Comments to the Author(s)

Attached (Appendix A).

Review form: Reviewer 2**Is the manuscript scientifically sound in its present form?**

Yes

Are the interpretations and conclusions justified by the results?

Yes

Is the language acceptable?

Yes

Do you have any ethical concerns with this paper?

No

Have you any concerns about statistical analyses in this paper?

No

Recommendation?

Major revision is needed (please make suggestions in comments)

Comments to the Author(s)

Arnot et al investigated whether male-female cohabitation and sexual frequency were associated with menopause onset. The study, based on data from Women's Health Across the Nation show that male-female cohabitation was not associated with menopause onset, however women who engage in more sex enter menopause late.

Major Comments

Did author have information on alcohol intake and economic status/occupation status? Alcohol and income is associated with menopause onset and should be included as covariate (Schoenaker et al; PE Teneri 2016).

Author select sexual frequency as the maximum amount of sexual activity in any of the categories: sexual touching/caressing; oral sex and sexual intercourse. It would be of interest to investigate the frequency within each strata and see whether the finding are independent of the type of sex. Also, it is not clear whether masturbation was included as sexual activity; it would be of interest to examine also whether masturbation is associated with menopause onset.

Did authors have information on frequency in orgasm since biologically these could be more relevant to menopause onset?

Being married is not associated with later onset of menopause in the univariate model, however after adjustment being married results at increased risk of early menopause. Is sexual frequency

driving this change? Sexual frequency could be a mediator in the association between being married and menopause onset, which could explain also the findings reported in the study.

Authors group married women and in a relationship in a single group, however there might be differences between a married couple and in a relationship. Could authors investigate the associations independently for married women and in a relationship?

I would recommend to rerun the analysis looking at the age of menopause as categorical variable: early vs. late menopause. This type of analysis would provide more meaningful results and easier to interpret.

Minor comment

The abstract in the current version is not very informative. Authors should add more information on number of women included in the study, follow-up time as well as estimates when reporting an association

Decision letter (RSOS-191020.R0)

04-Oct-2019

Dear Professor Arnot,

The editors assigned to your paper ("Sexual frequency is associated with age of natural menopause: results from the Study of Women's Health Across the Nation") have now received comments from reviewers. We would like you to revise your paper in accordance with the referee and Associate Editor suggestions which can be found below (not including confidential reports to the Editor). Please note this decision does not guarantee eventual acceptance.

Please submit a copy of your revised paper before 27-Oct-2019. Please note that the revision deadline will expire at 00.00am on this date. If we do not hear from you within this time then it will be assumed that the paper has been withdrawn. In exceptional circumstances, extensions may be possible if agreed with the Editorial Office in advance. We do not allow multiple rounds of revision so we urge you to make every effort to fully address all of the comments at this stage. If deemed necessary by the Editors, your manuscript will be sent back to one or more of the original reviewers for assessment. If the original reviewers are not available, we may invite new reviewers.

- Data accessibility

If you wish to submit your supporting data or code to Dryad (<http://datadryad.org/>), or modify your current submission to dryad, please use the following link:
<http://datadryad.org/submit?journalID=RSOS&manu=RSOS-191020>

- Competing interests

- Authors' contributions

- Acknowledgements

- Funding statement

on behalf of Professor Joris Veltman (Associate Editor) and Kevin Padian (Subject Editor)
openscience@royalsociety.org

Associate Editor's comments (Professor Joris Veltman):

With apologies for the unusual delay in completing review (we struggled a little to find suitable reviewers), we recommend revising the paper in-line with the comments of the reviewers who have been able to comment on the paper. Please bear in mind that you may not be granted a further round of revision, so please ensure you take all the comments of the reviewers into account in the revision - it would help enormously if you could provide not only a marked-up version of the revised m/s but also a point-by-point response to the reviewers with that revision. Good luck!

Comments to Author:

Reviewers' Comments to Author:

Reviewer: 1

Comments to the Author(s)

Attached

Reviewer: 2

Comments to the Author(s)

Arnot et al investigated whether male-female cohabitation and sexual frequency were associated with menopause onset. The study, based on data from Women's Health Across the Nation show that male-female cohabitation was not associated with menopause onset, however women who engage in more sex enter menopause late.

Major Comments

Did author have information on alcohol intake and economic status/occupation status? Alcohol and income is associated with menopause onset and should be included as covariate (Schoenaker et al; PE Teneri 2016).

Author select sexual frequency as the maximum amount of sexual activity in any of the categories: sexual touching/caressing; oral sex and sexual intercourse. It would be of interest to investigate the frequency within each strata and see whether the findings are independent of the type of sex. Also, it is not clear whether masturbation was included as sexual activity; it would be of interest to examine also whether masturbation is associated with menopause onset.

Did authors have information on frequency in orgasm since biologically these could be more relevant to menopause onset?

Being married is not associated with later onset of menopause in the univariate model, however after adjustment being married results at increased risk of early menopause. Is sexual frequency driving this change? Sexual frequency could be a mediator in the association between being married and menopause onset, which could explain also the findings reported in the study.

Authors group married women and in a relationship in a single group, however there might be differences between a married couple and in a relationship. Could authors investigate the associations independently for married women and in a relationship?

I would recommend to rerun the analysis looking at the age of menopause as categorical variable: early vs. late menopause. This type of analysis would provide more meaningful results and easier to interpret.

Minor comment

The abstract in the current version is not very informative. Authors should add more information on number of women included in the study, follow-up time as well as estimates when reporting an association

Author's Response to Decision Letter for (RSOS-191020.R0)

See Appendix B.

RSOS-191020.R1 (Revision)

Review form: Reviewer 2

Is the manuscript scientifically sound in its present form?

Yes

Are the interpretations and conclusions justified by the results?

Yes

Is the language acceptable?

Yes

Do you have any ethical concerns with this paper?

No

Have you any concerns about statistical analyses in this paper?

No

Recommendation?

Accept as is

Comments to the Author(s)

No further comments

Decision letter (RSOS-191020.R1)

02-Dec-2019

Dear Professor Arnot,

It is a pleasure to accept your manuscript entitled "Sexual frequency is associated with age of natural menopause: results from the Study of Women's Health Across the Nation" in its current form for publication in Royal Society Open Science. The comments of the reviewer(s) who reviewed your manuscript are included at the foot of this letter.

on behalf of the Associate Editor, and Professor Kevin Padian (Subject Editor)
openscience@royalsociety.org

Reviewer comments to Author:

Reviewer: 2

Comments to the Author(s)

No further comments

Appendix A

Sexual frequency is associated with age of natural menopause: results from the Study of Women's Health Across the Nation.

This paper tests the hypothesis that sexual frequency might explain the observed association between marital status and age at natural menopause (ANM). It also tests the theory that male pheromones influence ANM. The authors use a rich US data set which includes biomarkers and is specifically designed to investigate menopause. The methods are appropriate and the write-up is clear. This study is appropriate for RS Open Science readership and would make an important contribution to the literature. My main concern is the operationalisation of sex frequency and I would like to see a few robustness checks along these lines.

My specific comments are:

Page 2, line 15. Please add a short explanation for why exposure to male pheromones might influence ANM. This will also help clarify the predictions made by the two hypotheses given next.

Page 3, line 16. If support is found for either hypothesis, would this not suggest that the timing of menopause is facultative? Why only for H1?

Page 3, study sample section – please state if this study is nationally representative of the US. And if not, please add something in the discussion about generalisability/biases of the findings.

Page 4, the authors state: “Sexual activity other than intercourse was used to create the sex index as the hypothesis is predicting that cues from sex will result in a trade-off, and the underlying mechanism of sexual touching, oral sex, and masturbation could all signal possible pregnancy to the body.”

To make an adaptive argument, it may be that only sexual intercourse influences ANM (i.e. not touching, oral, or masturbation), especially if the mechanism has something to do with sperm in the reproductive tract – are the results still the same if you use only this outcome in the sex index?

Also, masturbation may signal exactly the opposite – that there is no male physical contact or likelihood of conception. Are your findings still robust if you just remove masturbation (i.e. keeping touching and oral as these include sexual partners)?

It is not clear why the frequency matters so much, having sex once a month or daily might both signal the possibility to conceive. Please add something to motivate this decision.

Do your findings have implications for the Grandmother Hypothesis (GH)? It may be that ceasing reproduction becomes adaptive when daughters/children start reproducing. Are you able to control for having grandchildren here? Even if not, please can you add a line or two in the discussion about this relationship (perhaps where you mention investment in existing kin; page 10, line 50). I realise that ceasing reproduction usually happens long before menopause but the GH is relevant to your adaptive argument.

Appendix B

Response to reviewers

Reviewer 1.

This paper tests the hypothesis that sexual frequency might explain the observed association between marital status and age at natural menopause (ANM). It also tests the theory that male pheromones influence ANM. The authors use a rich US data set which includes biomarkers and is specifically designed to investigate menopause. The methods are appropriate and the write-up is clear. This study is appropriate for RS Open Science readership and would make an important contribution to the literature. My main concern is the operationalisation of sex frequency and I would like to see a few robustness checks along these lines.

My specific comments are:

1. Page 2, line 15. Please add a short explanation for why exposure to male pheromones might influence ANM. This will also help clarify the predictions made by the two hypotheses given next.

RESPONSE: We have added in a short explanation into Section 1 to clarify why the authors of the pheromone hypothesis thought that it may be male cohabitation that delayed the menopause:

“One existing hypothesis postulates that the relationship between ANM and marital status is the result of male-female cohabitation [7]. This is based on the idea that exposure to male pheromones may increase the likelihood of having a regular menstrual cycle [24], with regular menstrual cycles having been observed to delay the menopause [25].”

2. Page 3, line 16. If support is found for either hypothesis, would this not suggest that the timing of menopause is facultative? Why only for H1?

RESPONSE: Thank you for bringing this up. The original pheromone hypothesis is not necessarily suggesting that there is something adaptive underlying the relationship.

Rather, it states that:

“... presence of a male in the household may affect, through pheromones, the hormonal milieu of the ovary, ultimately influencing age of menopause... How the presence of a male could be able to bring about a later age at menopause may be explained by the increased likelihood of regular menstrual cycles lengths of 29.5 (± 3 days) associated with male pheromones (Cutler et al., 1986b).” (Sievert et al., 2001:484).

We have clarified in text that ours is an alternative adaptive explanation, with sex signalling to the body the possibility of being pregnant, through saying:

“As an alternative, we propose an adaptive explanation based on energetic trade-offs”.

- Page 3, study sample section – please state if this study is nationally representative of the US. And if not, please add something in the discussion about generalisability/biases of the findings.

RESPONSE: Thank you for highlighting this. While SWAN is a community-based sample, existing literature has stated that SWAN is the largest, most diverse, and most representative study currently available to research aspects of the menopausal transition (Bromberger & Kravitz, 2011). We have added this information to Section 2.1.

- Page 4, the authors state: “Sexual activity other than intercourse was used to create the sex index as the hypothesis is predicting that cues from sex will result in a tradeoff, and the underlying mechanism of sexual touching, oral sex, and masturbation could all signal possible pregnancy to the body.” To make an adaptive argument, it may be that only sexual intercourse influences ANM (i.e. not touching, oral, or masturbation), especially if the mechanism has something to do with sperm in the reproductive tract – are the results still the same if you use only this outcome in the sex index?

RESPONSE: This is an interesting point. When originally discussing the hypothesis, we did think that – should the physical cue of sex be an indicator of possible pregnancy – other forms of sex would likely signal the same thing. As a result, prior to deriving the sex index for analysis, we checked whether the different forms of sex had the same relationship with ANM. Below are the results:

		HR (95% CI)	p
Intercourse (ref.: Less than monthly)			
	Monthly	0.82 (0.72-0.94)	<0.001
	Weekly	0.80 (0.71-0.91)	
Oral sex (ref.: Less than monthly)			
	Monthly	0.84 (0.74-0.95)	0.006
	Weekly	0.82 (0.69-0.97)	0.023
Sexual touching (ref.: Less than monthly)			
	Monthly	0.94 (0.81-1.08)	0.363
	Weekly	0.79 (0.71-0.89)	<0.001
Masturbation (ref.: Less than monthly)			
	Monthly	0.82 (0.70-0.96)	0.011
	Weekly	0.61 (0.49-0.77)	<0.001

As you can see, the hazard ratios follow the same direction, which is why we felt we were justified taking the maximum amount of sex experienced in any category as the 'sex index', as it appears that it is likely the same mechanism underlying the relationship between the individual categories and ANM.

5. Do your findings have implications for the Grandmother Hypothesis (GH)? It may be that ceasing reproduction becomes adaptive when daughters/children start reproducing. Are you able to control for having grandchildren here? Even if not, please can you add a line or two in the discussion about this relationship (perhaps where you mention investment in existing kin; page 10, line 50). I realise that ceasing reproduction usually happens long before menopause but the GH is relevant to your adaptive argument.

RESPONSE: SWAN unfortunately does not have data on grandchildren. The closest we have is self-reported data on who lives in their household; however, very few women within the sample live with their grandchildren.

We have added in a short statement about the links between our hypothesis and the Grandmother Hypothesis as follows on page 10:

"The idea that women cease fertility in order to invest in kin is known as the Grandmother Hypothesis, which predicts that the menopause originally evolved in humans to reduce reproductive conflict between different generations of females, and allow women to increase their inclusive fitness through investing in their grandchildren [30, 69, 70]. It may be costly for a woman to cease ovulatory function if the chances of her becoming pregnant are still high. In other words, if she is better able to increase her direct fitness, then it may be better to maintain the function of her menstrual cycle for slightly longer."

However, we are hesitant to draw too many parallels as the Grandmother Hypothesis is relevant to the *evolution* of the actual menopause, and very different physiological/evolutionary forces may be at play in regards to the cross-cultural variation in menopause *timing*.

Reviewer 2.

1. Did author have information on alcohol intake and economic status/occupation status? Alcohol and income is associated with menopause onset and should be included as covariate (Schoenaker et al; PE Teneri 2016).

RESPONSE: Thank you for this suggestion, and it is an important point. Yes, SWAN does have data on these things. However, there were a number of reasons we decided not to include them when selecting covariates.

Firstly addressing economic status/occupational status:

- i. Occupational status is only available at the baseline wave. We could have taken this variable as “occupation at baseline”, however, women have different ages at entry to the study, and also it is likely that a proportion of the women’s occupations would have changed throughout the study. For these reasons, we do not think an “occupation at baseline” variable would be particularly meaningful;
- ii. Employment status (measured as in paid employment vs. not in paid employment) is asked at most waves, so we could have included that. However, as this is a binary variable, we thought that other variables (e.g. education, where there are more categories) were better at capturing socioeconomic position;
- iii. Income was also available, however, this is categorised by SWAN into very broad income categories (see below), and therefore, once again we did not think it was terribly informative.

As a result, we chose to use education as a proxy of income/economic status. Furthermore, we show below that income and employment status are strongly associated with maximum educational attainment, and therefore we are hopefully demonstrating the effects of employment/occupation on ANM through using this variable:

Relationship between income and education at age of menopause: ($\chi^2 = p < 0.001$)

	Less than high school	High school	Some college	College degree	Post-graduate education
Less than \$19,000	28.21%	29.23%	28.21%	9.74%	4.62%
\$20,000-\$49,999	6.3%	18.7%	37.2%	19.29%	18.5%
\$50,000-\$99,999	2.51%	20.04%	34.66%	18.37%	24.43%
\$100,000 or more	1.49%	9.78%	25.37%	27.2%	36.15%

Relationship between employment status and education at age of menopause: ($\chi^2 = p < 0.001$)

	Less than high school	High school	Some college	College degree	Post-graduate education
Not employed	12.36%	24.14%	34.48%	17.82%	11.21%
Employed for pay	4.74%	15.73%	31.01%	19.21%	29.31%

We have similar reservations about including alcohol within the model. Firstly, within this dataset, alcohol consumption only appears to moderately increase the likelihood of entering menopause earlier when conducting a univariate analysis:

	HR (95% CI)	p
Frequency of alcohol consumption (ref.: Less than monthly)		
Monthly	1.12 (1.00-1.28)	0.047
Weekly	1.15 (0.95-1.40)	0.151

There is also a great deal of missingness in responses to the alcohol questions, which we determined not to be missing at random both due to the pattern of missingness, and existing literature stating that there is likely to be a bias in non-respondents towards those who drink alcohol more often. Furthermore, as noted in Taneri et al., (2016:525): *“The underlying mechanisms linking alcohol consumption to the time of onset of menopause are unknown”*, and they go on to explain that alcohol intake may act as a proxy for lifestyle habits, such as diet and physical activity. Furthermore, alcohol consumption is associated with changes in hormones which themselves associate with menopause timing. As we already have many measures of lifestyle, such as smoking, education attainment, BMI, and self-perceived health, we do not think it would be conducive to include alcohol within the model. Furthermore, as we control for estradiol, we are also hopefully accounting for the endocrine changes related to alcohol consumption. We also observe alcohol being significantly associated with many of the lifestyle indicators, meaning that we are likely already capturing any effects the variable would bring to the final model.

2. Author select sexual frequency as the maximum amount of sexual activity in any of the categories: sexual touching/caressing; oral sex and sexual intercourse. It would be of interest to investigate the frequency within each strata and see whether the finding are independent of the type of sex. Also, it is not clear whether masturbation was included as sexual activity; it would be of interest to examine also whether masturbation is associated with menopause onset.

RESPONSE: Thank you for pointing out it was unclear whether masturbation was included in the sex index. This has been clarified in text as follows:

“Due to the small amount of responses in some of the categories, the aforementioned measures of sex were aggregated into three new categories”.

Each measure of sex was similarly associated with menopause onset, with the results from univariate Cox regression presented below:

		HR (95% CI)	p
Intercourse (ref.: Less than monthly)			
	Monthly	0.82 (0.72-0.94)	<0.001
	Weekly	0.80 (0.71-0.91)	
Oral sex (ref.: Less than monthly)			
	Monthly	0.84 (0.74-0.95)	0.006
	Weekly	0.82 (0.69-0.97)	0.023
Sexual touching (ref.: Less than monthly)			
	Monthly	0.94 (0.81-1.08)	0.363
	Weekly	0.79 (0.71-0.89)	<0.001
Masturbation (ref.: Less than monthly)			
	Monthly	0.82 (0.70-0.96)	0.011
	Weekly	0.61 (0.49-0.77)	<0.001

As stated in text, the primary reason for aggregating the measures is that some of the categories had large amounts of missingness, and therefore – as all of them demonstrated a similar relationship with age of menopause – it made sense to take the ‘maximum sexual frequency’ as their sex index.

3. Did authors have information on frequency in orgasm since biologically these could be more relevant to menopause onset?

RESPONSE: This is an interesting thought. Having looked at the variable individually and how it relates to age of menopause, it does not appear to relate to age of menopause as demonstrated below:

		HR (95% CI)	p
Frequency of reaching climax (ref.: Less than monthly)			
	Monthly	0.91 (0.80-1.03)	0.145
	Weekly	1.10 (0.98-1.23)	0.125

Due to this, we have not included it within the paper, and think this further solidifies the idea that ANM may be related to the physical action of sex as a cue of possible pregnancy.

4. Being married is not associated with later onset of menopause in the univariate model, however after adjustment being married results at increased risk of early menopause. Is sexual frequency driving this change? Sexual frequency could be a mediator in the association between being married and menopause onset, which could explain also the findings reported in the study.

RESPONSE: Yes, this is correct. As we say in the abstract: “*We test an original hypothesis that sexual frequency acts as a bio-behavioural mediation between marital status and ANM*”.

5. Authors group married women and in a relationship in a single group, however there might be differences between a married couple and in a relationship. Could authors investigate the associations independently for married women and in a relationship?

RESPONSE: Thank you for this suggestion. We chose to group married women with women who are in a relationship as SWAN itself groups women who are married and living as married, which is already merging the boundary of in a relationship vs. married. However, we did create two new variables to test this, which were 'relationship status' (Not in a relationship; In a relationship) and 'marital status' (Single/Never married; Divorced/Separated; Married/Living as married; Widowed), with the results shown below:

	HR (95% CI)	p
(Variable used in paper) Marital status (ref.: Divorced/Separated/Single)		
Married/In a relationship	0.95 (0.84-1.08)	0.462
Widowed	1.13 (0.83-1.54)	0.448
Marital status (ref.: Divorced/Separated)		
Single/Never married	1.23 (1.00-1.50)	0.051
Married/Living as married	1.06 (0.94-1.26)	0.481
Widowed	1.11 (0.91-1.35)	0.295
Relationship status (ref.: Single)		
In a relationship	1.01 (0.92-1.11)	0.824

As can be seen, in univariate analyses, the new 'marital status' and 'relationship status' variables show that women who are married and in a relationship are more likely to enter menopause earlier (albeit to an insignificant degree), which are the results we see following adjustment in our current study.

We would like to retain the variable that we previously used in the manuscript, as in present day North America, there is a flexible marriage system, and sexual behaviour not only being confined to within a marriage, which we point out in the discussion on page 10. However, we do highlight the difference in coding decisions between studies and the influence that this can have on the results on pages 9 and 10, where we state:

“Conflicting results regarding marital status’ effect on ANM have been found elsewhere [e.g. 65], and one reason for this may be the way in which the researcher chooses to code the variable. In this analysis, romantic partnerships that may not have been acknowledged in previous studies due to having not been formalised by a marriage ceremony (e.g. cohabiting but unmarried) were taken into account. In addition, some prior studies have not included marital status as time-variant, and dichotomised the variable as ‘ever married’ or ‘never married’ [e.g. 66]. Hence, the fluid way in which this study coded marital status may account for the difference in results”.

6. I would recommend to rerun the analysis looking at the age of menopause as categorical variable: early vs. late menopause. This type of analysis would provide more meaningful results and easier to interpret.

RESPONSE: As this is an event history analysis, and women are coded as either having experienced menopause (1) or not experienced menopause (0), this suggestion is not possible. We agree that had this been some kind of general linear model, then a binary outcome would ease the interpretation of results (although often binary variables can cause the loss of information). However, this model essentially calculates the risk (hazard ratio) or a woman experiencing the event (menopause) at any given age, and therefore a categorical version of age of menopause is not valid for this form of analysis.

7. The abstract in the current version is not very informative. Authors should add more information on number of women included in the study, follow-up time as well as estimates when reporting an association

RESPONSE: Thank you for pointing this out. We have included a baseline sample size and primary effect size within the abstract.